# Cascade Dual-decoders Network for Abdominal Organs Segmentation

Ershuai Wang[1][0000−0003−4415−7643], Yaliang Zhao[1], and Yajun Wu[1]

Department of Research and Development, ShenZhen Yorktal DMIT Co. LTD
wuyj@yorktal.com

**Abstract.** In order to make full use of unlabeled images, we developed a pseudo-label based localization-to-segmentation framework for efficient abdominal organs segmentation. To reduce the target region, we locate the abdomen by a U-Net, then we train a fine organ segmentation model, which reduce the maximum usage of RAM memory. Segmentation with Dual-decoders is designed to improve the stability and cross supervise each other by pseudo labels. We also propose a class-weighted loss to pay more attention on the small organs like gallbladder, pancreas, which improve the mean DSC. Finally, we test the models on the public validation set, the total running time for the 50 CT images is 6676 seconds, the mean DSC is 0.8830 and the mean NSD is 0.9189.

**Keywords:** U-Net · Segmentation · Semi-Supervised Learning.

## 1 Introduction

In recent years, deep supervised learning methods have made excellent achievements in computer vision, especially in computer-aided diagnosis [2], such as lesion detection [3], tumor benign and malignant diagnosis [4], organ segmentation [6] and so on. Abdominal organ segmentation involves organ quantification, surgical planning, disease diagnosis and so on. On the one hand, there are many organs, including liver, kidneys, pancreas, spleen, stomach and other organs [8]. Each organ has different sizes and shapes, for example, the shape of stomach in different time varies a lot even for the same person, these make accurate pixel segmentation very difficult. On the other hand, manual labeling is expensive and time consuming. Besides, labeling medical images requires professional medical knowledge and rich experience, which makes it much more difficult to achieve the needs of practical application by using supervised learning method. Therefore, semi-supervised learning, which makes effective use of a large number of unlabeled data and less labeled data, has become a research hotspot in the field of deep learning.

This paper proposes a cascade abdominal organ segmentation model follow the semi-supervised learning. Our framework based on the famous nnU-Net [5], and we trained a prime model using the 50 labeled CTs with the default parameters. To expand the training dataset [10], we generate the pseudo labels for 2000 unlabeled training cases using the trained model. Then we develop a

cascade coarse-to-fine framework based on the provided labels and the pseudo labels. The first coarse model aims to obtain the rough location of the abdominal regions and the second fine model aims to segment the organs correctly. The fine model adopts cross pseudo training method [1,9], which reduces the feature noise influence and improves the stability.

The main contributions of this work are summarized as follows:

– We propose a cascade coarse-to-fine framework to make a trade-off of resource and precision.
– We design a dual-decoder model based on nn-UNet to make full use of unlabeled examples.
– We propose a class-weighted loss to improve the DSC of small organs.

## 2   Method

To expand the training dataset, we train a prime segmentation network based on nnU-Net with 50 provided labels first, then generate the pseudo labels for the unlabeled images. Finally, we train our coarse-to-fine framework using the all images and fine-tuning the segmentation model with the class-weighted loss. In this work, we do not optimize the segmentation efficiency of cascade framework.

### 2.1   Preprocessing

The preprocessing method in this paper refers to the fingerprint features of dataset proposed by nnU-Net [5], including the following steps:

– CT scans shape normalization   According to the average voxel spacing distance of the training data, the nearest neighbor interpolation is performed on the training data, that is, the CT's voxel spacing are rescale to 1.93×1.50×1.50 mm for localization model and 1×0.78×0.78mm for segmentation model.
– Voxel intensity normalization   The mean value, variance and values of 0.5% and 99.5% of all training samples are counted, then the voxel intensity is normalized. Concretely, the voxel intensity is truncated to [-973, 295], then minus the mean value 79.492 and divide the variance 142.997.

### 2.2   Pseudo Labeling

The objective of pseudo-labeling is to generate proxy labels to enhance the learning process [11,12]. Pseudo-labeling was successfully applied to a variety of tasks, such as image classification, semantic segmentation , text classification, machine translation and when learning from noisy data [12]. Therefore, we adopt pseudo labeling to enhance the abdominal organs segmentation.

To our best knowledge, nnU-Net has very good segmentation performance on many tasks even if we use the default parameters, such as number of layers,

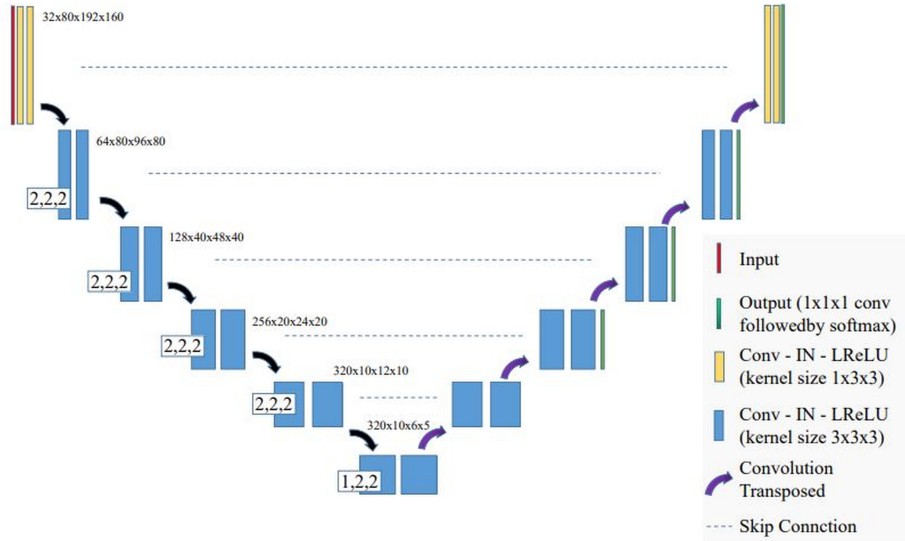

**Fig. 1.** Basic nnU-Net

number of filters per layer, number of pooling layers. So, we train a prime nnU-Net model with the provided 50 labeled cases at first. Concretely, the model has 5 stages in encoder and decoder as showed in Fig. 1, and there are 2 convolution layers in each stage. The input patch-size is $80\times192\times160$, the number of filters are [32, 64, 128, 256, 320], the stride of first 4 stages and the last stage are [2, 2, 2] and [1, 2, 2] respectively.

After 2000 epochs training, we test the model on the pubic validation set which got a 0.8671 mean DSC. Therefore, we believe that this model can segment the abdominal organs well. Then, we generate the 2000 unlabeled training cases on 2 computers with this model. In this way, we expand the training set a lot. However, we find that this model takes about very large RAM especially for cases imaging whole body. Also, this basic model is very slow, it takes about 5 days to generate 1000 cases.

### 2.3 Proposed Method

To reduce the memory usage and inference time, we propose a cascade framework. As shown in the Fig.2, the cascade framework consists of a localization model and a segmentation model, where the first localization model is used to determine the region of interest(ROI) before employing a segmentation model based on nn-UNet.

In a word, we locate the abdomen in low resolution space, then we segment the ROI in high resolution space.

**Fig. 2.** Cascade Framework

**Localization Model** Our localization model is implemented as a coarse binary segmentation U-Net where all labeled organs are treated as the foreground label and which is trained using the full image content after greatly downsampling the raw image. The localization model can generate binary segmentation for each input image, which are used to compute the bounding box of abdominal region which we define as the ROI. The network architecture was adapted from the U-Net and trained using the generalized dice loss, i.e.,

$$L_l = L_{GD}(p, t) \tag{1}$$

where $L_{GD}$ represents dice loss, $p$ and $t$ represent the predict label and the ground truth label. Our input size is 96×160×160, spacing are 1.93×1.50×1.50mm.

**Segmentation Model** The proposed segmentation network consists of one encoder and two decoders, as shown in Fig.3. Each encoding block is composed of two **Conv->BN->LReLU** sequences, as shown in Fig.4. Each decoding block consists of a up-sampling layer and an encoding block. We concatenate the feature channels between the decoder and the encoder with the same shape to reuse features and improve the ability of network feature extraction.

Specifically, the encoder includes five stages, in which the stride of the first convolution block of the coding block in the first and the last stage are [1, 2, 2], the stride for the other stages are [2, 2, 2], and the number of filters are [32, 64, 128, 256, 320]. The parameters for decoders are similar to the encoder but in the reverse direction. In this paper, the convolution kernel size are fixed as 3×3×3.

For inference, we can use both decoders to predict which behaves like ensemble to get a better segmentation. Alternatively, we can use any one of the decoders to predict for saving time. During this competition, we only use the first decoder branch.

**Loss Function** The training objective contains two parts: supervision loss and cross pseudo supervision loss:

$$L = l_s + \lambda l_{cps} \tag{2}$$

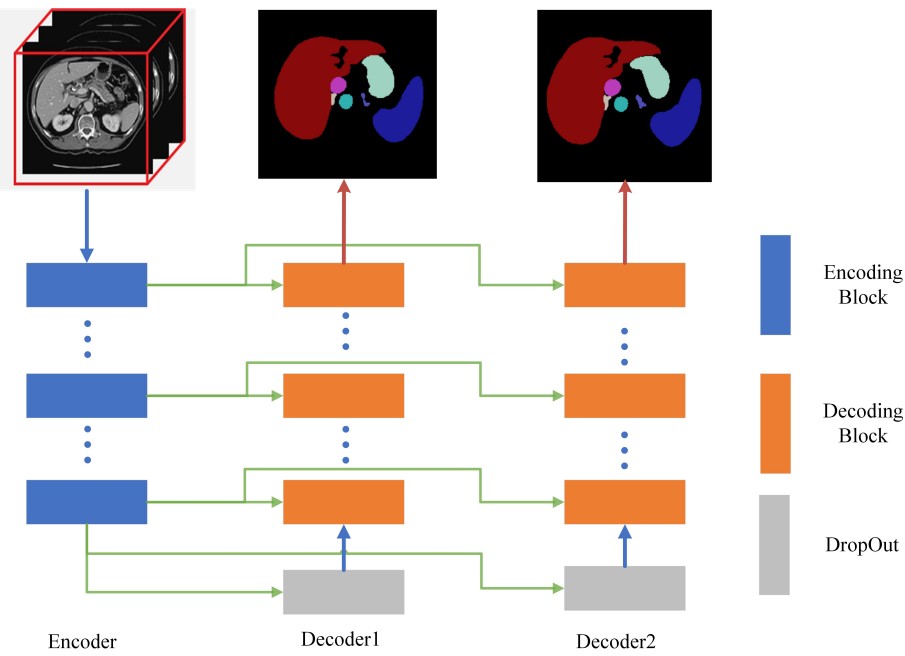

**Fig. 3.** Schematic of Segmentation Network

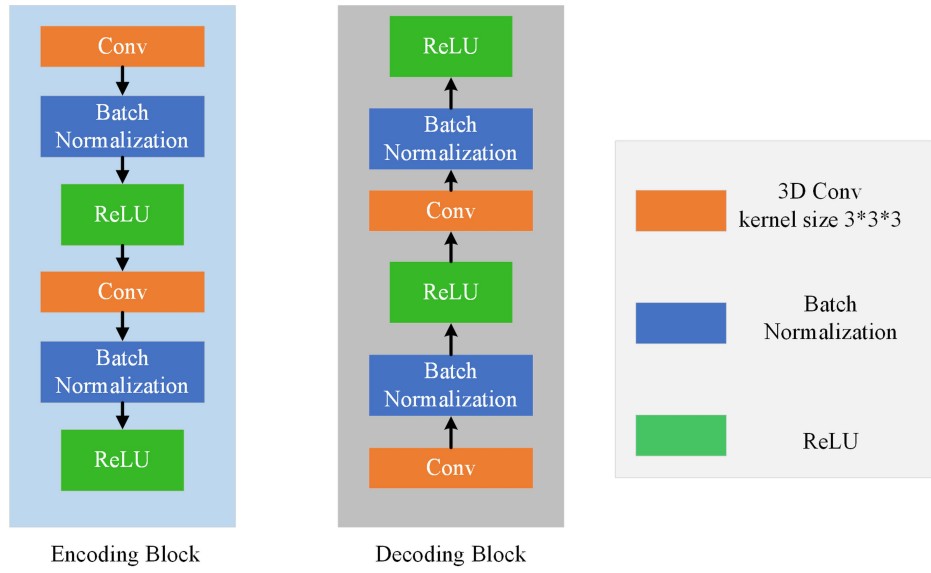

**Fig. 4.** Encoding and Decoding blocks

where $\lambda$ is the trade-off weight, we set $\lambda = 0.25$. The supervision loss $l_s$ is formulated using the standard pixel-wise cross-entropy loss $l_{cs}$ and dice similarity coefficient $l_{dsc}$ on the training CT scans over the two decoder paths:

$$l_s = \frac{1}{|T|} \sum_T \frac{1}{DHW} \sum_{i=1}^{DHW} \sum_{k=0}^{1} l_{ce}(p_{i,k}, t_i, w_c) + w l_{dsc}(\hat{t}_{i,k}, t_i, w_c) \qquad (3)$$

where $T$ is the training dataset consists of manual labeled part and pseudo label part, $l_{ce}$ is the cross-entropy loss function, $p_{i,k}$ is the predicted probability of the $k_{th}$ decoder, $t_i$ is the label, $\hat{t}_{i,k}$ is generate label, $w_c$ is the class weight based on the previous dice similarity coefficient and $w = 1$ is the trade-off weight between cross entropy and dice similarity. Specifically, the higher the dice score of a category, the more accurate the prediction of that category is. Therefore, more attention should be paid to the prediction of the other categories in subsequent training. Based on this, we adopt the form of dice reciprocal as the original class weight, then we normalized the weights by the sum. In the training process, the class weights are updated in a way similar to momentum optimization.

$$w_{ci} = \frac{1}{dsc_i + \epsilon} \qquad (4)$$

$$w_{ci} = \frac{w_{ci}}{\sum_{i=1}^{13} w_{ci}} \qquad (5)$$

$$w_{ci}^k = 0.5 w_{ci}^{k-1} + 0.5 w_{ci}^k \qquad (6)$$

where $dsc_i$ is the dice score of the $i-th$ organ, $\epsilon=$1e-6 is added to avoid the error of division by zero, $w_*^k$ is the weight of the $k-th$ epoch and we set $w_{ci}^0 = \frac{1}{13}$.

### 2.4   Post-processing

Connected component-based post-processing is commonly used in medical image segmentation. Especially in organ image segmentation, it often helps to eliminate the detection of spurious false positives by removing all but the largest connected component. So we compare the mean DSC between the prime predict and the post-processing. We find that the post-processing improve the DSC for liver, kidneys, spleen and aorta but degrade the DSC for small organs, especially for pancreas, gallblader and esophagus. Finally, we decide to do post-processing only on the liver, kidneys, spleen and aorta, thus improve the mean DSC by 0.0008.

## 3   Experiments

### 3.1   Dataset and evaluation measures

**Dataset**   The FLARE 2022 is an extension of the FLARE 2021 [7] with more segmentation targets and more diverse abdomen CT scans. The FLARE2022

dataset is curated from more than 20 medical groups under the license permission, including MSD [10], KiTS [3, 4], AbdomenCT-1K [8], and TCIA [2]. The training set includes 50 labelled CT scans with pancreas disease and 2000 unlabelled CT scans with liver, kidney, spleen, or pancreas diseases. The validation set includes 50 CT scans with liver, kidney, spleen, or pancreas diseases.

The testing set includes 200 CT scans where 100 cases has liver, kidney, spleen, or pancreas diseases and the other 100 cases has uterine corpus endometrial, urothelial bladder, stomach, sarcomas, or ovarian diseases. All the CT scans only have image information and the center information is not available.

**Evaluation measures**  The evaluation measures consist of two accuracy measures: Dice Similarity Coefficient (DSC) and Normalized Surface Dice (NSD), and three running efficiency measures: running time, area under GPU memory-time curve, and area under CPU utilization-time curve. All measures will be used to compute the ranking. Moreover, the GPU memory consumption has a 2 GB tolerance.

### 3.2  Implementation details

**Data augmentation**  We run the augmentations on the fly and with associated probabilities to obtain a never ending stream of unique examples the same as nnU-Net [5]. Concretely, we apply rotation, scaling, mirror, Gaussian noise, brightness variation and contrast variation on the sampled patches.

**Environment settings**  The development environments and requirements are presented in Table 1.

**Table 1.** Development environments and requirements.

| | |
|---|---|
| Windows/Ubuntu version | Windows 10 pro |
| CPU | Intel(R) Core(TM) i7-10700kF CPU@3.80GHz |
| RAM | 16×4GB; 2.67MT/s |
| GPU (number and type) | One NVIDIA RTX 3090 24G |
| CUDA version | 11.1 |
| Programming language | Python 3.8 |
| Deep learning framework | Pytorch (Torch 1.10, torchvision 0.9.1) |
| Link to code | https://github.com/Shenzhen-Yorktal/flare22 |

**Training protocols**  In the training process, the batch size is 2 and 500 patches are randomly selected from the training set per epoch, the patch size is fixed as 56 * 160 * 192. For optimization, we train it for 1500 epochs using SGD with a learning rate of 0.01 and a momentum of 0.9. During training, the learning rate is annealed following the poly learning rate policy, where at each iteration, the base learning rate is multiplied by .

**Table 2.** Training protocols.

| | |
|---|---|
| Network initialization | "he" normal initialization |
| Batch size | 2 |
| Patch size | 56×160×192 |
| Total epochs | 1500 |
| Optimizer | SGD with nesterov momentum ($\mu = 0.99$) |
| Initial learning rate (lr) | 0.01 |
| Lr decay schedule | poly learning rate policy $lr = 0.01 * (1 - \frac{e}{m})^2$ |
| Training time | 104.5 hours |
| Number of model parameters | 41.22M |
| Number of flops | 59.32G |

## 4    Results and discussion

### 4.1    Quantitative results on validation set

We perform controlled experiments with the same training configurations as described in Section 3.2. As the baseline, we train a U-Net-like model with the 50 labeled data only. Then we generate the pseudo labels for the 2000 unlabeled data, and re-train the same model with the labeled data and pseudo labeled data. After that, we fine-tune the model with class-weighted loss. Finally, we obtain 3 models with different mean DSC: 0.8671, 0.8749, 0.8890.

Table 3 illustrates the detailed results on validation set. It's obvious that the pseudo label improves the baseline DSC for most organs except RAG and LK . We think that the pseudo label behaves like the augmentation, which enlarges the training set 40 times and improving the mean DSC. The degradation of RAG and LK come from the over-fitting of the baseline model. Class-weighted loss, assign larger weights for organs with poor DSC, improves the DSC a lot for RAG, LAG, gallbladder and LK. We think that the class-weighted loss is similar to the attention mechanism, which degrades the extreme high DSC slightly but improves the others.

We choose the class-weighted model for the FLARE22, and the all followings are based on this.

### 4.2    Qualitative results on validation set

Both DSC and NSD scores vary greatly in abdominal organs segmentation between different case. For example, the mean DSC for validation case21 and case48 are 0.967 and 0.686, the mean NSD are 0.995 and 0.716.

Fig.5 presents some well-segmented and challenging cases in the validation set. It can be observed that for the well-segmented cases, the predictions are almost the same with the ground truths. We think that the satisfying segmentation come from the clear boundaries and good contrast of the organs. In contrast with

**Table 3.** Segmentation DSC of abdominal organs.

| DSC | labeled only | all | class-weighted loss |
|---|---|---|---|
| Liver | 0.9721 | 0.9807 | 0.9790 |
| Right Kidney(RK) | 0.9140 | 0.9257 | 0.9387 |
| Spleen | 0.9569 | 0.9727 | 0.9580 |
| Pancreas | 0.8505 | 0.8882 | 0.8701 |
| Aorta | 0.9560 | 0.9674 | 0.9601 |
| IVC | 0.8833 | 0.9026 | 0.9018 |
| RAG | 0.8367 | 0.8226 | 0.8603 |
| LAG | 0.8345 | 0.8367 | 0.8603 |
| Gallbladder | 0.7279 | 0.7401 | 0.7650 |
| Esophagus | 0.8263 | 0.8690 | 0.8754 |
| Stomach | 0.8673 | 0.8952 | 0.8941 |
| Duodenum | 0.7584 | 0.7764 | 0.7915 |
| Left Kidney(LK) | 0.8890 | 0.8722 | 0.9027 |
| **Mean** | **0.8671** | **0.8749** | **0.8890** |

the well-segmented cases, the challenging cases are poor, which missing some organs part or all, as shown in Fig.5(b). We think that the bad segmentation come from the heterogeneous lesions and the unclear boundaries.

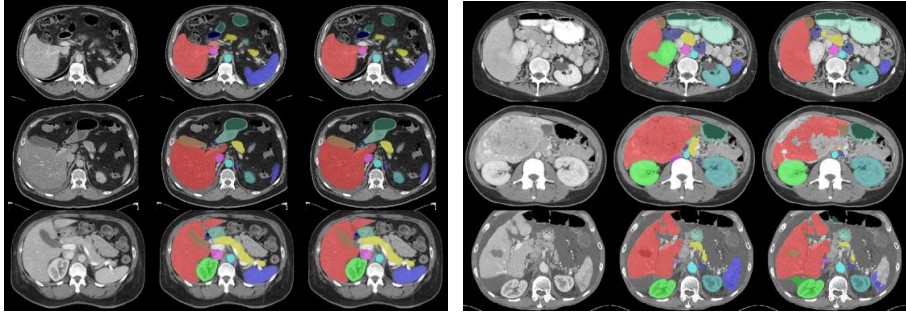

(a) Well-segmented cases          (b) Challenging cases

**Fig. 5.** Well-segmented and challenging cases from validation sets

### 4.3   Segmentation efficiency results on validation set

We run our models on a docker with NVIDIA 1080 GPU(12GB) and 64GB RAM for the 50 validation cases. The mean running time per case is 133.5 seconds, the maximum GPU memory used is 3041MB and the maximum RAM used is 27466MB. We find that the large RAM consumption are during the prediction of validation Case 10 and 50 which are scans of full body. The average AUC of GPU and CPU are 252158 and 2422 respectively, which is really high because of the long inference time.

### 4.4   Results on final testing set

According to the requirement, we submit the docker to FLARE22 and the organizer run the model on the hidden test set, which consists of 200 CT scans. The final mean DSC is 0.8981 and the mean NSD is 0.9367, which are close to the result on validation set. The detail showed in table 4. We can see that most organs have very high DSC except gallbladder, esophagus and duodenum. We think it due to the relatively smaller volume of these organs.

**Table 4.** Results on final testing set

|  | DSC | NSD |
| --- | --- | --- |
| Liver | 0.9820 ± 0.0137 | 0.9843±0.0288 |
| Right Kidney(RK) | 0.9488±0.1505 | 0.9466±0.1520 |
| Spleen | 0.9716±0.0618 | 0.9732±0.0751 |
| Pancreas | 0.8569±0.1257 | 0.9431±0.1257 |
| Aorta | 0.9649±0.0352 | 0.9811±0.0415 |
| IVC | 0.9063±0.0855 | 0.9063±0.0954 |
| RAG | 0.8934±0.0753 | 0.9768±0.0738 |
| LAG | 0.8774±0.1117 | 0.9612±0.1059 |
| Gallbladder | 0.8266±0.2903 | 0.8355±0.2988 |
| Esophagus | 0.8220±0.1530 | 0.9032 ±0.1511 |
| Stomach | 0.9133±0.1048 | 0.9317±0.1118 |
| Duodenum | 0.7741±0.1741 | 0.8978±0.1401 |
| Left Kidney(LK) | 0.9385±0.1506 | 0.9363±0.1411 |
| **Mean** | **0.8981** | **0.9367** |

### 4.5   Limitation and future work

As showed in section 4.3, our model use large RAM for some cases and the GPU memory used is higher than 2048 MB. Besides, the DSC of gallbladder and some tiny organs are much poorer than liver. Therefore, we will focus on the speed and the specified organ segmentation in the future.

## 5   Conclusion

During the training, we find that the unlabeled images improve the performance which proves the data-driven of deep learning again. And we use cross pseudo supervise to improve the model further, which shows the Semi-Supervised-Learning power in computer vision. It would also be interesting to adapt and examine the effectiveness of SSL in other visual tasks and learning settings.

**Acknowledgements** We declare that the segmentation method they implemented for participation in the FLARE 2022 challenge has not used any pre-trained models nor additional datasets other than those provided by the organizers. The proposed solution is fully automatic without any manual intervention.

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
