# OpenReview forum: "Cascade Dual-decoders Network for Abdominal Organs Segmentation"
_MICCAI.org/2022/Challenge/FLARE_

### Official Review · Reviewer_keDw · 2022-09-12
**The paper proposed an effective cascade dual-decoder network, supervised by cross pseudo supervision loss.**

**Rating:** 9
**Confidence:** 4

**Review:**

Pros:
1.The paper proposed a cascade network to localized and segmented the input data, which effectively reduce the influence of unrelated area.
2. By using dual decoder framwork, the performance was improved by using cross pseudo supervision loss. Thus, the information in unlabeled data were used in a better way, which is simple but effective.
3.Class-weighted loss was used to further improve the performance, especially for small organs.
4.Experiment results shown the effectiveness of the model.

Cons:
1.If equations of calculation process for class weight loss are given, it would be helpful for better understanding.
2.The training time and the computational cost were relatively high, which may worth further improvement.
3.The test result section was missing.

---

> ### Author Response · Authors · 2022-10-09
> **revision**
>
> Q1: If equations of calculation process for class weight loss are given, it would be helpful for better understanding.
> A1: I have add the equations of class weight at the end of the section 2.3
>
> Q2:The training time and the computational cost were relatively high, which may worth further improvement.
> A2: In this work, we focus on the accurancy, we will do some work for optimization in future.
>
> Q3:The test result section was missing.
> A3: I have add the final test result after i got the response from organizer.

---

### Official Review · Reviewer_HjR2 · 2022-09-15
**Great work but talk a little about semi-supervise method**

**Rating:** 8
**Confidence:** 3

**Review:**

* In method, it seem that author did not introduce or cite any paper of cross pseudo supervision loss. It makes me did not understand how author bring unlabeled data into training procedure.

The result is impressive but the method how author use unlabeled data is still unclear.

---

> ### Author Response · Authors · 2022-10-09
> **revision**
>
> Q1:In method, it seem that author did not introduce or cite any paper of cross pseudo supervision loss. It makes me did not understand how author bring unlabeled data into training procedure.
>
> A1:I  added the references and give a short introduce about how we use the unlabeled data in section 2.2.

---

### Official Review · Reviewer_8ssy · 2022-09-17
**MICCAI-FLARE**

**Rating:** 8
**Confidence:** 4

**Review:**


Advance:
1. Design a dual-decoder model based on nnUNet to make full use of unlabeled examples, and get a hight score.
2. The semi-supervised method improved by 0.021.

Weakness:
1. They didn’t optimize the segmentation efficiency.
2. Lack of algorithm time acceleration analysis.

---

> ### Author Response · Authors · 2022-10-09
> **Revision**
>
> Q1:They didn’t optimize the segmentation efficiency.
> Q2:Lack of algorithm time acceleration analysis.
>
> A: In this work, we focus on the accurancy, we will do some work on optimization in future.

---

### Official Review · Reviewer_SWfv · 2022-09-18
**In this paper, the coarse-to-fine segmentation framework and loss function are introduced in detail. It is recommended that the authors explain the pseudo label generation process in detail, and analyze the reasons for poor cases.**

**Rating:** 6
**Confidence:** 3

**Review:**

This paper introduces the coarse-to-fine segmentation framework, which can locate and then segment organs. In this paper, a dual-decoders model based on nnUNet is proposed, which uses the pseudo label to enhance the segmentation performance. In addition, a class-weighted loss function is proposed to enhance the segmentation effect for small organs.

Suggested Improvements:
This paper can introduce pseudo-label generation in detail, including the selection of network model and parameter setting in the nnUNet framework. This paper did not analyze the reasons for bad cases. In the section of segmentation efficiency results, the AUC(GPU and CPU) index can be displayed.

---

> ### Author Response · Authors · 2022-10-09
> **Revision**
>
> Q1:This paper can introduce pseudo-label generation in detail.
> A1: Thanks. I added section 2.2 for pseudo labeling.
>
> Q2:This paper did not analyze the reasons for bad cases.
> A2: I tried to explain the reason for the bad cases and added it at the end of section 4.2. We think it is caused by the lessions and the unclear boundaries.
>
> Q3:In the section of segmentation efficiency results, the AUC(GPU and CPU) index can be displayed.
> A3: Thanks. I added the AUC at the end of section 4.3

---

### Official Review · Reviewer_YmKm · 2022-09-20
**Great framework design for this two-phase network**

**Rating:** 8
**Confidence:** 4

**Review:**

Pros: Great two-phase method and framework design, which seems to bring great performance
Cons: The efficiency of infer needs to be improved

---

> ### Author Response · Authors · 2022-10-09
> **Revision**
>
> Q1: The efficiency of infer needs to be improved.
>
> A1: Thansk. We will do some work in the future.

---

### Official Review · Reviewer_k5oc · 2022-09-21
**Good work!**

**Rating:** 8
**Confidence:** 4

**Review:**

In this work, they proposed a segmentation method containing two stages: localization and segmentation. The final mean DSC is 0.88 and the average inference time is 133.5s. The RAM usage and GPU consumption are low.

---

> ### Author Response · Authors · 2022-10-09
> **Revision**
>
> Thanks.

---

### Public Comment · ~Zhengshan_Huang1 · 2022-09-21
**The article is clearly structured and contains all the necessary content**

The article is clearly structured and contains all the necessary content

---

> ### Author Response · Authors · 2022-10-09
> **Revision**
>
> Thanks!

---

### Meta-Review · Program_Chairs · 2022-09-28

**Recommendation:** Minor Revision
**Confidence:** 5

**Metareview:**

Nice paper. Please address the reviewers' comments in the revised manuscript.